# Short-Term Exposure to Fine Particulate Matter and Hospitalizations for Acute Lower Respiratory Infection in Korean Children: A Time-Series Study in Seven Metropolitan Cities

**DOI:** 10.3390/ijerph18010144

**Published:** 2020-12-28

**Authors:** Jongmin Oh, Changwoo Han, Dong-Wook Lee, Yoonyoung Jang, Yoon-Jung Choi, Hyun Joo Bae, Soontae Kim, Eunhee Ha, Yun-Chul Hong, Youn-Hee Lim

**Affiliations:** 1Department of Occupational and Environmental Medicine, School of Medicine, Ewha Womans University, Seoul 120-750, Korea; Jongminoh@ewha.ac.kr (J.O.); eunheeha@ewha.ac.kr (E.H.); 2Department of Preventive Medicine, Chungnam National University College of Medicine, Daejeon 301-747, Korea; cwohan@cnu.ac.kr; 3Department of Preventive Medicine, Seoul National University College of Medicine, Seoul 110-799, Korea; is2uz@snu.ac.kr (D.-W.L.); yuyujang88@snu.ac.kr (Y.J.); hierica8@snu.ac.kr (Y.-J.C.); 4Environmental Health Center, Seoul National University College of Medicine, Seoul 03080, Korea; 5Korea Environment Institute, Sejong 30147, Korea; hjbae@kei.re.kr; 6Department of Environmental and Safety Engineering, Ajou University, Suwon 16499, Korea; soontaekim@ajou.ac.kr; 7Department of Public Health, University of Copenhagen, 1599 Copenhagen, Denmark; limyounhee@gmail.com

**Keywords:** fine particulate matter, acute lower respiratory infection, time-series, children

## Abstract

Although several studies have evaluated the association between fine particulate matter (PM_2.5_) and acute lower respiratory infection (ALRI) in children, their results were inconsistent Therefore, we aimed to evaluate the association between short-term exposure to PM_2.5_ and ALRI hospitalizations in children (0–5 years) living in seven metropolitan cities of Korea. The ALRI hospitalization data of children living in seven metropolitan cities of Korea from 2008 to 2016 was acquired from a customized database constructed based on National Health Insurance data. The time-series data in a generalized additive model were used to evaluate the relationship between ALRI hospitalization and 7-day moving average PM_2.5_ exposure after adjusting for apparent temperature, day of the week, and time trends. We performed a meta-analysis using a two-stage design method. The estimates for each city were pooled to generate an average estimate of the associations. The average PM_2.5_ concentration in 7 metropolitan cities was 29.0 μg/m^3^ and a total of 713,588 ALRI hospitalizations were observed during the 9-year study period. A strong linear association was observed between PM_2.5_ and ALRI hospitalization. A 10 μg/m^3^ increase in the 7-day moving average of PM_2.5_ was associated with a 1.20% (95% CI: 0.71, 1.71) increase in ALRI hospitalization. While we found similar estimates in a stratified analysis by sex, we observed stronger estimates of the association in the warm season (1.71%, 95% CI: 0.94, 2.48) compared to the cold season (0.31%, 95% CI: −0.51, 1.13). In the two-pollutant models, the PM_2.5_ effect adjusted by SO_2_ was attenuated more than in the single pollutant model. Our results suggest a positive association between PM_2.5_ exposure and ALRI hospitalizations in Korean children, particularly in the warm season. The children need to refrain from going out on days when PM_2.5_ is high.

## 1. Introduction

Many studies have reported that fine particulate matter (PM_2.5_) is associated with respiratory morbidity and the mortality of children [1,2,3]. Children’s respiratory systems can be more susceptible to PM_2.5_ due to their fast breathing rate, narrow bronchi, and immature immune system [4,5]. Exposure to PM_2.5_ in children has been linked to respiratory diseases such as asthma, allergic inflammation, and acute lower respiratory infection (ALRI) [6,7,8,9].

ALRI is an ordinary respiratory disease in young children, and its clinical symptoms are characterized by fever, shortness of breath, sore throat, and cough. Acute bronchitis and bronchiolitis, influenza, and pneumonia are included in ALRI [10]. Globally, ALRI is the fifth major cause of child death, accounting for 30% to 50% of child hospitalizations [7,11].

The Global Burden of Disease group suggested that air pollution, including PM_2.5_, is a risk factor for ALRI in children [7,11,12]. Respiratory infections in children or newborns can be fatal, and they also put a considerable burden on parents. Some previous studies reported evidence supporting the biological plausibility of an association between PM_2.5_ and ALRI. PM_2.5_ is comprised of diverse heavy metals and organic components and may persuade oxidative stress and inflammation in the lung tissues [13,14]. Toxicological research studies have shown that PM_2.5_ exposure may lead to impaired immune responses by altering the phagocytic activities of alveolar macrophages and reducing antioxidant levels [7,15].

Previous studies showed inconsistent results regarding PM_2.5_ exposure and ALRI in children. Although several epidemiology studies in China, the US and Australia showed positive associations [1,10,16], some studies have found no association between PM_2.5_ and ALRI in children [17,18].

Korea has relatively high PM_2.5_ levels (annual mean concentration of 25 µg/m^3^ in 2016) among the countries of the Organization for Economic Co-operation and Development [19] and a high prevalence of pediatric ALRI hospitalization (20–30% prevalence) [20]. However, a recent Korean study showed no relationship between PM_2.5_ and nationwide childhood respiratory hospitalization in 2013–2015 [21]. Given the high concentration levels of PM_2.5_ and the prevalence of ALRI in metropolitan cities, additional studies of any association, particularly in urban areas, are urgently needed. In some previous studies, only certain cities were considered to be within the scope of the research area [16,18,22]. In other cases, the study period was relatively short [21]. The inconsistency of prior research may be due to differences in exposure by region, which may result in regional differences, and the short duration of the study may also be a factor. In this study, we investigated the relationship between PM_2.5_ and children’s ALRI hospitalization using the Korea’s National Health Insurance data over a longer period (2008–2016). The aim of our study was to evaluate the impact of short-term exposure to PM_2.5_ on ALRI in children. We also investigated sex- and season-specific estimates.

## 2. Methods

### 2.1. Number of Daily Children’s ALRI Hospitalization

In this study, we utilized the children’s ALRI hospitalization data archived from the National Health Insurance Database (NHID) in South Korea. The NHID contains all of the medical information of the insured persons, including hospital visits (date of admission, date of discharge, and diagnostic codes) and personal information (sex, birth year, and residential address). Detailed information on NHID and its customizing processes for researchers have been described elsewhere [23]. Since this data is big data, it can represent the characteristics of the population. The National Health Information Service (NHIS) prepares queries and provides data at the request of researchers. We claimed information on ALRI hospitalization for children living in seven metropolitan cities (Seoul, Busan, Daegu, Incheon, Daejeon, Gwangju, and Ulsan) of South Korea from 2008 to 2016 (Figure 1). The ALRI hospitalization was determined by the International Classification of Disease, 10th revision diagnostic codes: either primary or secondary diagnostic codes of J20–J22 (J20: Acute bronchitis, J21: Acute bronchiolitis, J22: Unspecified acute lower respiratory infection). Daily counts of hospitalization for ALRI were calculated in each metropolitan city.

This study uses the unidentifiable data prepared by the NHIS and follows the guidelines for the protection of personal information provided by the NHIS. Therefore, our research was exempted from deliberation by the Institutional Review Board (E-1807-038-956) of Seoul National University Hospital.

### 2.2. PM_2.5_ Modeling Data

Since the nationwide monitoring system for PM_2.5_ levels has only been in operation since 2015, we used the Community Multiscale Air Quality (CMAQ, version 4.7.1) modeling system to estimate PM_2.5_ concentrations in areas throughout the study period (2008–2016). These modeling data are designed by the US Environment Protection Agency. Several previous studies have used this modelled data [24,25]. A detailed description of the CMAQ PM_2.5_ data is given in Appendix A. Exposure values were estimated for each of 74 administrative districts in seven cities. We calculated the modeled data by administrative districts on a daily average by city.

### 2.3. Air Pollution Monitoring Data and Environmental Variables

We also collected data on other air pollutants to construct multiple pollutant models, considering that a single pollutant can be biased or misleading [26].

Information on PM_10_, sulfur dioxide (SO_2_), nitrogen dioxide (NO_2_), carbon monoxide (CO) and ozone (O_3_) was archived from the AirKorea database (https://www.airkorea.or.kr/web). This database includes hourly data collected from the stationary monitoring stations located in each metropolitan area (number of monitoring stations in 2018, Seoul: 25, Busan: 17–20, Daegu: 13, Incheon: 17, Gwangju: 7, Daejeon: 10, Ulsan: 16). Measurement methods for each air pollutant are well described in previous studies. [27]. Using the hourly data of air pollutants in multiple monitoring stations, we computed daily average values for each region.

We collected the daily environmental variables (mean temperature, relative humidity, dew point temperature etc.). These data were prepared from the Korea Meteorological Administration database. We can use the data for research purposes on this website (http://www.kma.go.kr), and we have obtained the data by each city. We wanted to simplify the model by using variables containing temperature, humidity and dew point information. So, we used daily mean apparent temperature (AT) [28,29]. The equation for AT is described in the Appendix A.

### 2.4. Statistical Analysis

We analyzed the time-series data for daily ALRI hospitalization count. A two-stage design was considered to investigate the relationship between PM_2.5_ and ALRI hospitalization. Briefly, we evaluated the generalized additive model (GAM) analysis to estimate city-specific associations. Then, we used random-effects meta-analysis to provide an effect estimate for the whole city.

In the first stage, we constructed a GAM to estimate the association between PM_2.5_ exposure and ALRI hospitalizations for each city. The daily ALRI hospitalization is count data and assumes a Poisson distribution in the model. Exposure-response curves in the metropolitan cities showed linear associations (Figure 2). Hereafter, we made an assumption that the association was linear between PM_2.5_ exposure and ALRI hospitalization. The detailed linear model is as follows:*Log*(*μ_t_*) = *Intercept* + *β* × *PM_2.5_* + *s*(*Time*, *df* = 7 × 9) +*s*(*AT_t_*, *df* = 6) + *γ* × *Day of week*(1)

The *μ_t_* is the daily ALRI hospitalization count in each metropolitan city on day *t* and *β* is the estimate of the association between PM_2.5_ and ALRI hospitalization, *AT_t_* is daily AT in a metropolitan city on day *t*, *Day of week (DOW)* indicates the categorical variable with a reference day of Sunday (from Monday to Sunday), *γ* means the estimate of the *DOW* and *s* is a penalized spline term. The spline function for *Time* was applied to control for unmeasured confounder (e.g., seasonality, trends) of daily ALRI hospitalization counts [30]. Since air pressure, wind speed and other meteorological variables did not improve the model, they were excluded from the main model. We estimated the associations of single-lag (lag 0, 1, …, 7) and moving average-lag (Lag 01, 02, …, 07) days of PM_2.5_ with ALRI hospitalization. To focus on the short-term effects of PM_2.5_ on ALRI hospitalization, we considered the following PM_2.5_ exposure windows: single-lag day exposure from current day (lag 0) to 7 days before hospitalization (lag 7) and moving average exposure by averaging PM_2.5_ levels for 2 days (current day and 1 previous day before hospitalization, lag 01) to 8 days (current day and 7 previous days before hospitalization, lag 07). The main results are evaluated as a percentage change (%) of the odds ratio of daily ALRI hospitalization associated with a 10 μg/m^3^ increase in 7-day moving average exposure to PM_2.5_. The percentage change was calculated using the following formula: $(e^{\beta *10}$ − 1) × 100.

To select degrees of freedom (*df*) for time trends, we performed the analysis by allocating the *df per year* from 1 to 9 (i.e., *df* = 1 × 9, …, 9 × 9) and evaluated the patterns of the effects estimates in each metropolitan city. The *df* for AT was selected by referring to previous papers [31].

We pooled the city-specific estimates of the associations between PM_2.5_ and ALRI hospitalizations in urban areas. We used the random effect model to account for the heterogeneity of the estimates across regions [32,33]. We evaluated the heterogeneity of the model through I^2^, H^2^ and Cochran’s Q statistics. If the I^2^ statistic is less than 25%, there is low heterogeneity, medium for 25 to 50%, and high heterogeneity for greater than 50% [34].

When the H^2^ statistic is close to 1, this indicates homogeneity [35]. The Cochran’s Q statistic is based on a chi-square distribution, and a larger value means greater variation across studies rather than within subjects within a study [36]. We presented the meta-analysis results in a forest plot.

Subgroup analyses were conducted by sex (boys and girls) and season (warm season (April to September) and cold season (October to March)).

### 2.5. Two-Pollutant Model

We constructed a two-pollutant model to control for the effect of other pollutants on the ALRI hospitalization [37] because risk estimates can be affected by the accompanying air pollutants in the two-pollutant model [26]. We used the Variance Inflation Factor (VIF) for each pollutant pair to evaluate the multi-collinearity between the two pollutants (i.e., PM_2.5_ and other pollutants). If the VIF value of the other pollutant was less than 10, we included the pollutant in the two-pollutant models [38].

The software used for all data pre-processing and analysis are SAS Ver. 9.4 (SAS Institute Inc., Cary, NC, USA) and R Programming Language (Ver. 4.0.0, R Foundation for Statistical Computing, Vienna, Austria).

## 3. Results

### 3.1. Main Results

From 2008 to 2016, there were 713,588 cases of children (0–5 years) hospitalized in seven metropolitan cities of South Korea (Seoul: 147,225, Busan: 213,336, Daegu: 90,329, Incheon: 60,606, Gwangju: 112,196, Daejeon: 34,944, and Ulsan: 54,952) (Table 1). The number of daily ALRI hospitalizations was slightly higher in boys (*n* = 397,333 (55.7%)) than in girls (*n* = 316,255 (44.3%)). The daily ALRI hospitalization distribution shows the shape of a right-skewed distribution (Appendix A). Table 2 describes the summary statistics for daily air pollutants and environmental variables. The average (standard deviation, SD) daily PM_2.5_ concentration during the 2008 to 2016 was 29.0 (7.6) μg/m^3^ (Gwangju being the lowest, 24.8 (7.9) μg/m^3^ and Seoul being the highest, 30.6 (8.3) μg/m^3^) (Figure 1 and Table 2). The average (SD) daily SO_2_, NO_2_ and O_3_ concentrations (ppm) were 0.005 (0.002), 0.024 (0.009) and 0.025 (0.010), respectively.

We found slightly greater estimations of the associations when we applied 1 to 4 *df* per year (Appendix A). Therefore, we selected *7 df* per year (63 *df* = 7 per year × 9 years) for *Time* (Appendix A). The exposure–response relationship between PM_2.5_ and the risk of ALRI hospitalizations are presented in Figure 2. In Figure 2, we showed a strong linear association between PM_2.5_ and the risk of ALRI hospitalization. Figure 3 shows the city-specific estimates and pooled estimates between PM_2.5_ exposure and ALRI hospitalization risk. A 7-day moving average PM_2.5_ was associated with a 1.20% (95% confidence interval (CI): 0.71, 1.71) increase in ALRI hospitalization risk.

### 3.2. Subgroup Analysis

In the stratified analysis by sex, we found a similar association of PM_2.5_ exposure with ALRI hospitalization (boys: 1.31%, 95% CI: 0.75, 1.86; girls: 1.02%, 95% CI: 0.45, 1.58). However, we also observed differences in the associations by the season. The association in the warm season was shown (warm season: 1.71%, 95% CI: 0.94, 2.48; cold season: 0.31%, 95% CI: −0.51, 1.13).

### 3.3. Two-Pollutants Model

There were positive correlations between the modeled PM_2.5_ and the monitored pollutants: PM_10_ [correlation coefficient r =  0.96], SO_2_ (r = 0.76), NO_2_ (r =  0.68), and CO (r = 0.75). However, there was no correlation between modeled PM_2.5_ and monitored O_3_ (r = −0.03) (Figure 4). In the two-pollutant model, we considered SO_2_, NO_2_ and O_3_. After adjusting for NO_2_ and O_3_, the model was similar to that of the single pollutant model (Table 3). However, after it was SO_2_ adjusted, the association with PM_2.5_ was unrelated (0.69%, 95% CI: −0.34, 1.74).

### 3.4. Sensitivity Analysis

The results for the single-day and moving average-day lag models are shown in Figure 5, Appendix A. In a single-day lag model, we observed decreasing estimates with increasing lag days (Lag 0: 0.58%, 95% CI: 0.41, 0.76; Lag 7: −0.01, 95% CI: −0.18, 0.16). In the moving average lag model, the estimated association with the 7-day moving average exposure to PM_2.5_ was the highest (Lag 06: 1.20%, 95% CI: 0.71, 1.71). The residual plots and histograms for our main model in the first stage showed no specific patterns (Appendix A). In an additional analysis to validate the CMAQ Modeled PM_2.5_ concentration levels, we observed higher correlation coefficients between the CMAQ modeled and monitored PM_2.5_ data levels in each of the seven cities (0.96 or higher) (Appendix A).

## 4. Discussion

We used the time series data for 9 years on children’s ALRI hospitalization. In our study, we evaluated the association between PM_2.5_ and ALRI hospitalization in urban areas of South Korea. We found a 1.2% increase in ALRI hospitalization per 10 μg/m^3^ increase in the 7-day moving average exposure to PM_2.5_. A strong linear association was observed between PM_2.5_ and ALRI hospitalizations, and the association was stronger in the warm season. However, we found no association in the cold season. The associations of PM_2.5_ with ALRI hospitalization were slightly attenuated after adjusting for NO_2_, O_3_, and SO_2_ in the two-pollutant model.

Previous short-term studies have evaluated the relationship between air pollution and ALRI in children and have mostly conducted their analyses with time-series data or in a case-crossover design. Time-series data for 94,315 children under age 15 in China revealed the association between PM_2.5_ and ALRI hospitalizations (≤1:1.26% (0.03, 2.50; 2–4 years: 1.24% (0.35, 2.14); 5–14 years old: 1.72% (0.34, 3.12)) [16]. In a study with time-series data from Brazil, increases of PM_10_ was associated with an increase in hospitalizations for pediatric or adolescent respiratory diseases, including upper respiratory tract acute infections, acute bronchitis, bronchiolitis, asthma, pneumonia, bronchopneumonia and other respiratory diseases [39]. In another case-crossover study with time-series data of 50,857 children under 3 years old in Brazil, a 4-day moving average PM_10_ was associated with a 4% increase in bronchiolitis hospitalizations [40]. In a case-crossover study for multi cities, exposures to PM_2.5_, PM_10_, NO_2_, and SO_2_ were associated with hospitalization of children with pneumonia and acute bronchitis [41]. In a case-crossover study with 112,467 children in the US, exposure to PM_2.5_ within a week was related to ALRI hospitalization aged 0–2 years, 3–17 years and over 18 years [10]. A Malaysia study evaluated a positive association between primary air pollutants (PM_10_, NO_2_ and O_3_) and ALRI hospitalization of children in cities [42].

A Vietnam study showed that exposure to PM_2.5_ increased ALRI hospitalization for children aged 0–4 (3.51%, 95% CI: 0.96, 6.12). In this time-series study, boys were higher risk of PM_2.5_ exposure (male: 5.22%, 95% CI: 0.86, 9.78; female: −5.84, −4.30) [22]. In our study, the subgroup analysis by sex was similar, but the risk for boys was slightly higher. This is probably because boys are more presumably exposed to ambient PM_2.5_ than girls [22]. In Korea, there was a study showing increased respiratory mortality in infants with exposure to high PM_10_ levels [43]. Another nationwide study in Korea in children aged 0-4 reported a stronger association between PM_2.5_ and ALRI hospitalization in the cold season [21]. The discrepancy between the two studies by Kim et al. and our study may come from the different definitions of ALRI hospitalization (primary diagnostic codes of ALRI in Kim et al. vs. primary or secondary diagnostic codes in our study), study periods (2013–2015 in Kim et al., vs. 2006–2015 in our study), study areas (nine provinces and seven metropolitan cities in Kim et al., vs. only seven metropolitan cities in our study), and analysis methods (a difference-in-differences approach in Kim et al., vs. a Poisson regression model in our study).

There are a few studies reporting contradictory results. A study with time-series data of US emergency room hospitalizations has revealed an association of PM_2.5_ with pneumonia and upper respiratory diseases, but not with bronchiolitis hospitalizations [17]. In a study with 17,118 children under 18 years old, the association between hospitalization due to pneumonia was associated with PM_2.5_, but bronchitis and asthma were not related to PM [44]. Another case-crossover study with 19,901 infants in California found no link between PM_2.5_ and bronchitis [18].

There were several previous studies considering the two-pollutants model. Zheng et al. showed that the estimate of the association with PM_2.5_ was not significant in the two-pollutant model after controlling for SO_2_ and NO_2_ (PM_2.5_: 1.50, 95% CI: 0.35, 2.66; PM_2.5_ adjusted for SO_2_: 0.17, 95% CI: −2.55, 0.43; PM_2.5_ adjusted for NO_2_: 0.17, 95% CI: −1.47, 1.85) [16]. Barnett et al. argued that PM_2.5_ and SO_2_ concentration levels are a proxy measure of the same emission source (e.g., motor vehicles), and they may impact each other [41]. They showed that SO_2_ attenuated the effects of PM_2.5_ in the adjusted two-pollutant model, which is similar to our findings.

Most air pollutants, including fine dust, are caused by traffic, industries and other sources of combustion [37,41]. These air pollutants are highly correlated because they are affected by climatic conditions (temperature, humidity, wind speed, air pressure, and etc.). In this study, air pollutants excluding ozone were highly correlated with PM_2.5_ (Figure 4). Therefore, it is important to note that models containing multiple air pollutants may have collinearity.

We observed a greater estimate of the link between PM_2.5_ and ALRI hospitalization in the warm season. The association between PM_2.5_ and ALRI in children aged 1–4 was more outstanding in warm seasons in Australia and New Zealand in a previous study, which is consistent with our study [41]. A high temperature is well-known to have undesirable health effects; a high temperature exacerbates pre-existing health conditions [45] and is related to the death of susceptible population group (e.g., young children, elderly) [46]. Exposure to PM_2.5_ is associated with decreased heart rate variability (HRV), suggesting it triggers a change in the autonomic nervous system [47,48]. Interestingly, exposure to high temperature was associated with decrease in HRV during the warm season [49]. This suggests that high levels of PM_2.5_ and ambient temperature could have interaction effects on autonomic nervous system [50], which could interact with the immune system [51]. On the other hand, another study in San Paulo showed greater effects of PM_10_ during the cold season [52]; however, the average ambient temperature in San Paulo occurs during the warm season, and 13–22 °C is considered the cold season, which is very different from the significant seasonal changes in ambient temperature in Korea. Discrepancies of the season-specific associations may come from the geographical and climatic characteristics of PM_2.5_ [52,53]. The composition of PM_2.5_ may vary by season, and different compositions of PM_2.5_ may cause various adverse effects, particularly in urban areas with heavy traffic volumes [54]. In cold seasons, while the concentration of fine dust is much higher than in warm seasons, the contributions of secondary sulfate salts and nitrate salts are greater during the summer in Beijing, China [55]. Although the PM_2.5_ concentration was also higher in the cold season in Seoul as well, the concentration of NH^4+^ was higher in the cold season while SO_4_^2−^ was higher during the warm season [56]. Different study periods, geographic coverages, and definitions of outcomes may also impact the study results. In addition, exposure to high temperatures and fine dust can affect the lung function of the elderly as well as in ALRI on children [57,58]. The interaction of high temperatures and PM_2.5_ may be the cause, and these effects need to be addressed in future studies.

Some studies have reported PM_2.5_ and ALRI relevance, explaining oxidative stress and inflammatory reactions as causal mechanisms. PM_2.5_ can cause oxidant stress to the lung tissue and induce systemic inflammation [59,60]. PM_2.5_ stimulates the expression of inflammatory cytokines, which may cause systematic inflammation [61].The PM_2.5_ inhaled in the body mainly affects macrophages and epithelial cells in the respiratory tract. In vitro and in vivo studies have described that PM_2.5_ reduces the phagocytic activities of alveolar macrophages, making people vulnerable to air pollutants [62,63]. Because of immature lung development, children are vulnerable to health effects if they inhale PM_2.5_. In addition, children breathe more air pollutants because they breathe faster than adults.

Young children may be diagnosed with ALRI for respiratory syncytial virus (RSV) infection and bronchitis. If children with RSV or bronchitis are exposed to PM_2.5_, the ALRI risk may increase [10].

This study has several limitations. First, we were unable to use monitored data of PM_2.5_ due to the short period of nationwide monitoring of PM_2.5_. So, we compared modeled PM_2.5_ levels with monitored data in 2015 and 2016 to validate the modeled concentration levels of PM_2.5_. We found a high correlation between modeled and monitored exposure levels in years with available data for both PM_2.5_ levels (2015 and 2016) (Appendix A). There are several studies that have estimated and evaluated the CMAQ PM_2.5_ concentrations [64,65]. These studies found that CMAQ PM_2.5_ concentrations can be affected by weather variables (temperature, wind, etc.), emissions inventory, etc. In addition, during the 2013–2014 period, CMAQ PM_2.5_ concentrations may be underestimated due to high concentrations of air pollution and high emissions [64]. High concentrations were included and emissions were high in 2013 and 2014, so that the reproducibility of the simulation was somewhat lower than that of other years. However, the level of model performance evaluation was not low, so we concluded that the CMAQ PM_2.5_ data were suitable for our study. Second, we were unable to gather individual-level data such as socioeconomic status, parental education levels, and secondary smoking status due to the health insurance nature of the data used in our study. Third, we could not control the unmeasured confounding effects [66]. However, the time trends in the model may control for short and long-term temporal fluctuations, which may reduce unmeasured confounding effects [67]. Fourth, exposure data were applied as the daily average in each region. Therefore, potential misclassification may have been caused by exposure values. Finally, this research was evaluated in metropolitan cities of South Korea. Thus, our results cannot be generalized to rural areas or other countries with different PM_2.5_ levels.

## 5. Conclusions

Our results have revealed a link between PM_2.5_ and ALRI hospitalization among Korean children. The increase in PM_2.5_ is related to the increased ALRI hospitalization for Korean children. In particular, exposure to PM_2.5_ in warm seasons was highly related to ALRI. Children need to refrain from outdoor activity when PM_2.5_ is high. The results of our research can be used for children’s health care and health policies for children’s respiratory disease. Future research needs to investigate the link between air pollution, temperature and children’s respiratory health.

Since PM_2.5_ is mainly caused by traffic and industrial emissions, government control is necessary. The South Korean government announced its 2nd Seoul Metropolitan Air Quality Control Master Plan from 2015 to 2024. It includes regulations such as reducing eco-friendly cars and carbon emissions and strengthening management around life. In the future, managing air pollution is expected to reduce children’s health effects caused by exposure to air pollution.

## Figures and Tables

**Figure 1 ijerph-18-00144-f001:**
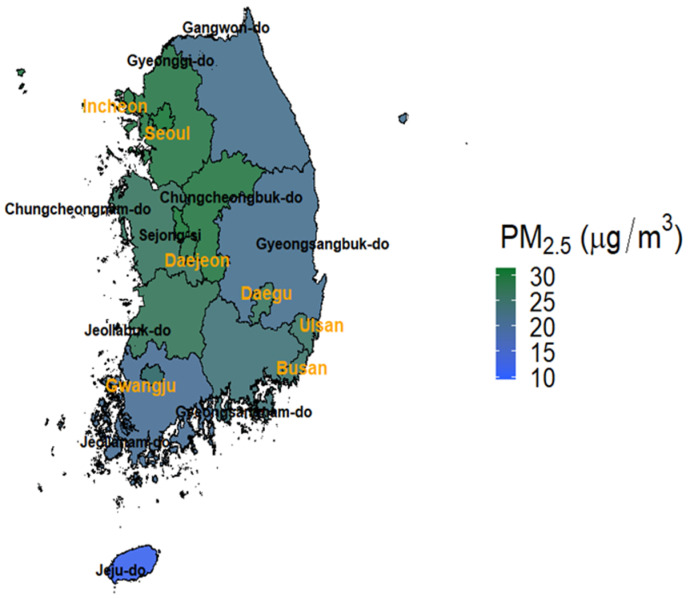
Map of Korea and average fine particulate matter (PM_2.5_) concentrations during the study period (2008–2016). The names of seven metropolitan cities are indicated with orange color.

**Figure 2 ijerph-18-00144-f002:**
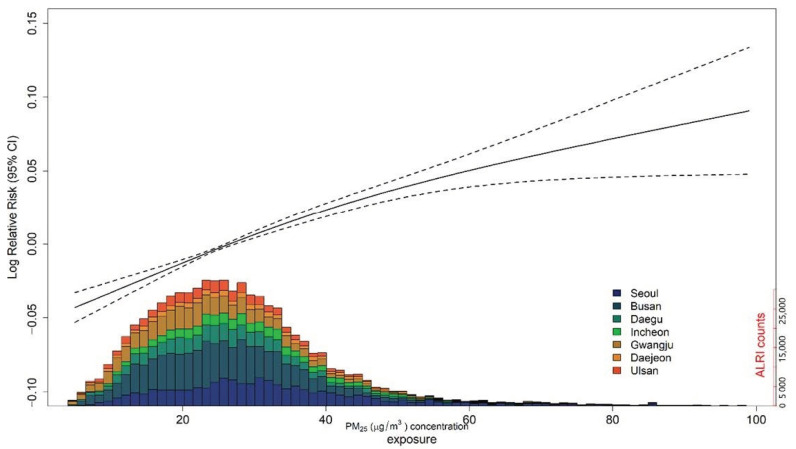
Exposure–response relationship between short-term PM_2.5_ concentrations and ALRI hospital admissions in children in seven metropolitan cities in Korea. The gray shade indicates the 95% confidence interval. The figure below shows the ALRI hospital admission distribution over PM_2.5_ concentrations. All models used the 7-day moving average exposure to PM_2.5_ μg/m^3^. The percentage change is calculated as the change per PM_2.5_ 10 μg/m^3^ increase.

**Figure 3 ijerph-18-00144-f003:**
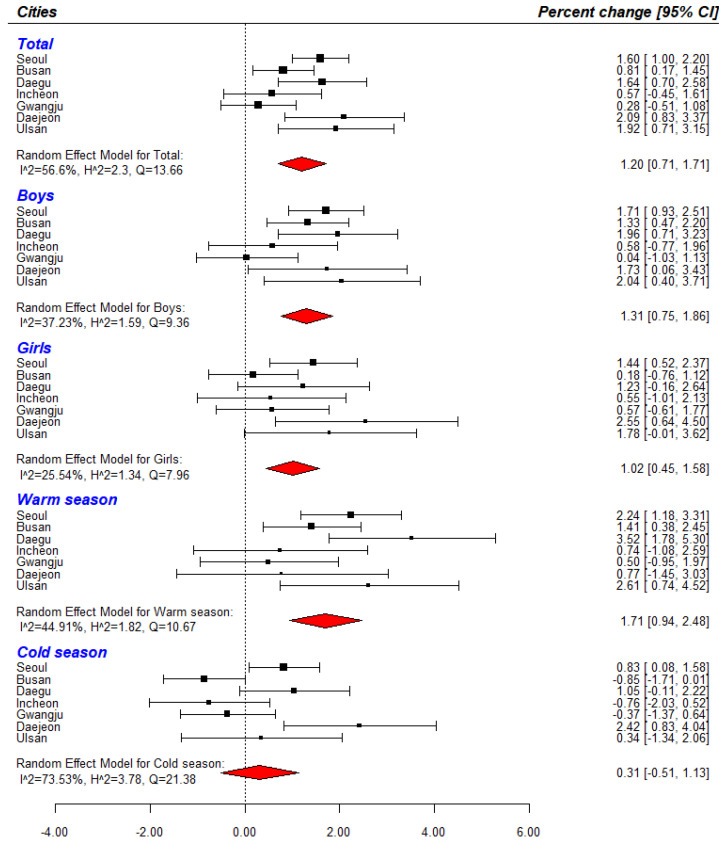
Meta-analysis of the association between PM_2.5_ and ALRI hospital admissions in seven major cities in Korea (Percentage change (%) per 10 μg/m^3^). All models were used the 7-day moving average exposure to PM_2.5_ μg/m^3^. Time-series model adjusted for same covariates (time, AT, day of weeks)

**Figure 4 ijerph-18-00144-f004:**
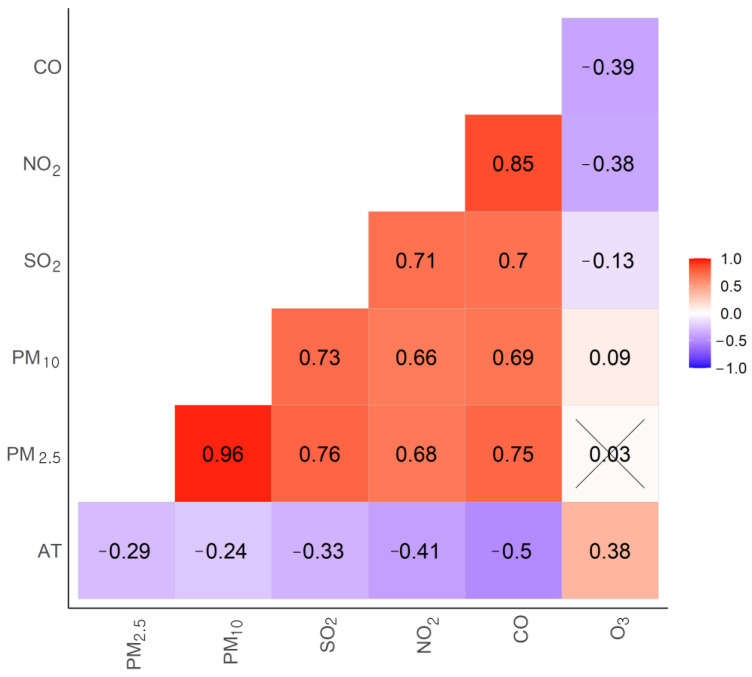
Spearman’s correlation analysis of modeled PM_2.5_ and monitored air pollutants PM_10_, SO_2_, NO_2_, CO, and O_3_) and apparent temperature during the study period. All correlation coefficients were statistically significant (*p*-value < 0.05). Statistically insignificant correlations are indicated by X-marks.

**Figure 5 ijerph-18-00144-f005:**
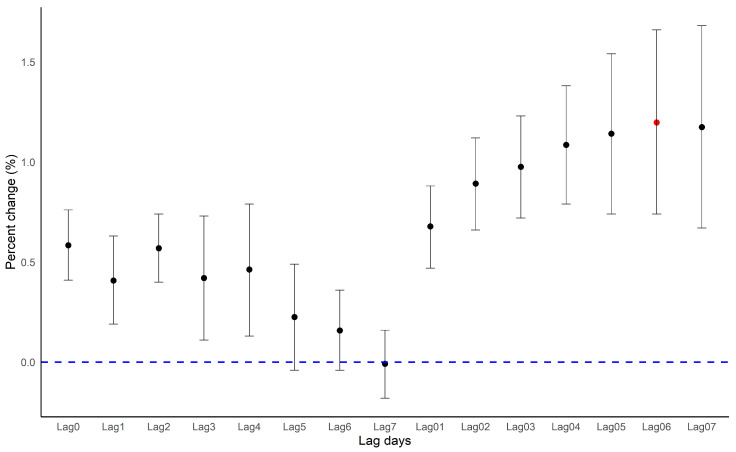
Percentage change (%) for ALRI hospitalizations following single lag days (lag 0 to lag 7) and moving average (lag 01 to lag 07) PM_2.5_ exposure in the meta-analysis. Time-series model adjusted for the same covariates (time, AT, day of weeks). The percentage change is calculated as the change per PM_2.5_ 10 μg/m^3^ increase. The blue line means the border line. The red circle means the highest percentage change.

**Table 1 ijerph-18-00144-t001:** Number of acute lower respiratory infection (ALRI) hospitalizations by sex and season for the seven metropolitan cities of Korea from 2008 to 2016. The warm season is April–September and the cold season is January–March, October–December.

Category	Total	Seoul	Busan	Daegu	Incheon	Gwangju	Daejeon	Ulsan
Sex								
Boys	397,333	19,855	85,099	116,538	61,456	34,358	49,815	30,212
Girls	316,255	15,059	62,126	96,798	50,740	26,248	40,514	24,740
Seasons								
Warm	334,766	66,075	102,548	41,269	29,095	52,887	16,566	26,326
Cold	378,822	81,150	110,788	49,060	31,511	59,309	18,378	28,626
Total	713,588	147,225	213,336	90,329	60,606	112,196	34,944	54,952

**Table 2 ijerph-18-00144-t002:** Descriptive statistics for daily air pollutants and environmental variables in seven metropolitan cities in Korea, 2008–2016. The warm season is April–September and the cold season is January–March, October–December.

Category	Total	Season		Cities						
		Warm	Cold	Seoul	Busan	Daegu	Incheon	Gwangju	Daejeon	Ulsan
PM_2.5_ (μg/m^3^)	29.0 (7.6)	24.2 (13.4)	30.1 (17.2)	30.6 (8.3)	25.3 (6.1)	26.4 (6.9)	29.1 (8.3)	24.8 (7.9)	28.6 (9.2)	25.6 (6.2)
PM_10_ (μg/m^3^)	46.9 (23.7)	43.1 (23.9)	50.6 (28.5)	47.7 (29.4)	47.3 (24.0)	47.0 (25.7)	52.6 (28.9)	43.0 (26.7)	43.1 (24.8)	47.4 (24.9)
SO_2_ (ppm)	0.005 (0.002)	0.005 (0.003)	0.006 (0.002)	0.005 (0.002)	0.006 (0.002)	0.004 (0.002)	0.007 (0.002)	0.004 (0.001)	0.004 (0.002)	0.008 (0.003)
NO_2_ (ppm)	0.024 (0.009)	0.021 (0.009)	0.028 (0.011)	0.033 (0.012)	0.021 (0.007)	0.023 (0.011)	0.028 (0.012)	0.020 (0.008)	0.021 (0.009)	0.023 (0.008)
CO (ppm)	0.500 (0.163)	0.419 (0.114)	0.581 (0.221)	0.544 (0.210)	0.398 (0.108)	0.481 (0.202)	0.588 (0.218)	0.501 (0.180)	0.492 (0.212)	0.494 (0.141)
O_3_ (ppm)	0.025 (0.010)	0.031 (0.011)	0.019 (0.009)	0.021 (0.011)	0.028 (0.010)	0.025 (0.012)	0.024 (0.011)	0.027 (0.012)	0.024 (0.013)	0.026 (0.010)
Mean apparent temperature (°C)	13.9 (11.5)	22.8 (7.7)	4.9 (6.8)	12.7 (11.9)	15.2 (10.2)	14.5 (11.2)	12.7 (12.2)	14.3 (11.9)	13.1 (12.2)	14.5 (10.7)
Mean temperature (°C)	13.9 (9.7)	21.4 (5.2)	6.3 (6.9)	12.8 (10.8)	15.1 (8.3)	14.7 (9.7)	12.6 (10.1)	14.4 (9.6)	13.2 (10.2)	14.4 (8.8)
Mean relative humidity (%)	64.6 (16.7)	69.7 (15.3)	59.5 (16.4)	60.2 (14.9)	62.4 (18.3)	57.8 (16.9)	71.7 (16.2)	67.5 (13.7)	67.8 (14.4)	64.9 (17.6)
Mean dew point (°C)	6.5 (11.6)	14.9 (7.3)	−1.9 (8.7)	4.5 (12.1)	7.2 (11.6)	5.3 (11.8)	7.1 (11.8)	7.7 (10.7)	6.6 (11.4)	7.1 (11.6)

**Table 3 ijerph-18-00144-t003:** Percentage change in PM_2.5_ single pollutant model and two-pollutant model. The percentage change is calculated as the change per PM_2.5_ 10 μg/m^3^ increase. All models used the 7-day moving average exposure to PM_2.5_ μg/m^3^. The time-series model was adjusted for the same covariates (time, AT, day of weeks).

Cities	Single Pollutant (95% C.I)	Two-Pollutant (95% C.I)		
	PM_2.5_	PM_2.5_ + SO_2_	PM_2.5_ + NO_2_	PM_2.5_ + O_3_
Seoul	1.60 (1.00, 2.20)	1.58 (0.80, 2.37)	1.84 (1.09, 2.60)	1.56 (0.96, 2.16)
Busan	0.81 (0.17, 1.45)	−0.96 (−1.71, −0.19)	0.31 (−0.48, 1.10)	0.94 (0.28, 1.60)
Daegu	1.64 (0.70, 2.58)	1.60 (0.45, 2.77)	1.15 (0.00. 2.32)	1.55 (0.60, 2.50)
Incheon	0.57 (−0.45, 1.61)	−0.98 (−2.26. 0.32)	0.62 (−0.61, 1.87)	0.45 (−0.59, 1.50)
Gwangju	0.28 (−0.51, 1.08)	−0.09 (−1.04, 0.86)	0.14 (−0.79, 1.08)	−0.01 (−0.84, 0.83)
Daejeon	2.09 (0.83, 3.37)	2.70 (1.09, 4.34)	3.01 (1.55, 4.49)	1.74 (0.44, 3.05)
Ulsan	1.92 (0.71, 3.15)	1.45 (0.00, 2.92)	1.40 (−0.16, 2.99)	1.91 (0.66, 3.17)
Total	1.20 (0.71, 1.71)	0.69 (−0.34, 1.74)	1.14 (0.43, 1.85)	1.11 (0.60, 1.64)

## Data Availability

The data used in this study is available from the NHIS. However, this data requires deliberation by the NHIS and analysis is conducted in a limited closed network. Only limited analysis results can be obtained.

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
