# Peer review of "Short-Term Exposure to Fine Particulate Matter and Hospitalizations for Acute Lower Respiratory Infection in Korean Children: A Time-Series Study in Seven Metropolitan Cities"

_ijerph, 2020, doi:10.3390/ijerph18010144_

Round 1

Reviewer 1 Report

The manuscript is great with additional insights of what is going in the area of air pollution and the lung health of the children in S Korea.

However, the authors need to address the potentials confounders especially given the fact that they did not have individual-level data such SES and others. How do you overcome that limitation?

They need to provide convincing pathophysiological mechanisms behind PM2.5 and ALRI in children, more in line with children's vulnerability and susceptibility to air pollution.

Could there be interaction between PM2.5 and other air pollutants? The reader will need to know where these air pollutants come from, are they from industries? automobiles? or both?, or transboundary sources? 

And then, finally, the authors need to provide more specific recommendations regarding the control of air pollution using the best examples from other countries. 

The manuscript will need a review from a native English speaker or someone with excellent command of English.

Author Response

Major Comment 1

“The manuscript is great with additional insights of what is going in the area of air pollution and the lung health of the children in S Korea.

However, the authors need to address the potentials confounders especially given the fact that they did not have individual-level data such SES and others. How do you overcome that limitation?”

Response

Thank you for your comment. The time-series data used in our study are daily ALRI hospitalization counts. To see the relationship between PM2.5 and ALRI, time series data analysis focuses on population, not individuals.

As you point out, for individual level data, indicators such as secondhand smoke (SHS) are potential confounding in children’s lung health.

These individual indicators cannot be adjusted in the analysis of time-series data on the relationship between air pollution and health effects. Therefore, if the study design is an individual-based design, confounding such as SHS will be considered in the model.

--------------------------------------------------------------------------------------------

Major Comment 2

“They need to provide convincing pathophysiological mechanisms behind PM2.5 and ALRI in children, more in line with children's vulnerability and susceptibility to air pollution.”

Response

Thank you for your comment. We agree with your suggestion. We added a biological mechanism in which PM2.5 exposure affects children’s acute lower respiratory infections in the Discussion section.

After revision (In the Discussion section )

“Some studies have reported PM2.5 and ALRI relevance, explaining oxidative stress and inflammatory reactions as causal mechanisms. PM2.5 can cause oxidant stress to the lung tissue and induce systemic inflammation [1, 2]. PM2.5 stimulates the expression of inflammatory cytokines, which may cause systematic inflammation [3].The PM2.5 inhaled in the body mainly affects macrophages and epithelial cells in the respiratory tract. In vitro and in vivo studies have described that PM2.5 reduces the phagocytic activities of alveolar macrophages, making people vulnerable to air pollutants [4, 5]. Because of immature lung development, children are vulnerable to health effects if they inhale PM2.5. Also, children breathe more air pollutants because they breathe faster than adults.

Young children may be diagnosed with ALRI for respiratory syncytial virus (RSV) infection and bronchitis. If children with RSV or bronchitis are exposed to PM2.5, the ALRI risk may increase [6].”

--------------------------------------------------------------------------------------------

Major Comment 3 

“Could there be interaction between PM2.5 and other air pollutants? The reader will need to know where these air pollutants come from, are they from industries? automobiles? or both?, or transboundary sources?”

Response

Thank you for your comment. We agree with your suggestion. As you point out, air pollutants, including PM2.5, have various causes of generation. We added the contents related to the Discussion section.

Before revision

None

After revision (In the Discussion section)

" Most air pollutants, including fine dust, are caused by traffic, industries and other sources of combustion [7, 8]. These air pollutants are highly correlated because they are affected by climatic conditions (temperature, humidity, wind speed, air pressure and etc.). In this study, air pollutants excluding ozone were highly correlated with PM2.5. Therefore, it is important to note that models containing multiple air pollutants may have collinearity.”

--------------------------------------------------------------------------------------------

Major Comment 4

“And then, finally, the authors need to provide more specific recommendations regarding the control of air pollution using the best examples from other countries.”

Response

Thank you for your comment. We agree with your suggestion. In the Conclusion section, we added recommendations on air pollution prevention and comments on future work. But, the recommendations on air pollution control were made around Korea, not other countries.

After revision (In the Conclusion section)

“Our results have revealed a link between PM2.5 and ALRI hospitalization among Korean children. The increase in PM2.5 is related to the increased ALRI hospitalization for Korean children. In particular, exposure to PM2.5 in warm seasons was highly related to ALRI. Children need to refrain from outdoor activity when PM2.5 is high. The results of our research can be used for children’s health care and health policies for children’s respiratory disease. Future research needs to investigate the link between air pollution, temperature and children's respiratory health.

Since PM2.5 is mainly caused by traffic and industrial emissions, government control is necessary. The South Korean government announced its 2nd Seoul Metropolitan Air Quality Control Master Plan from 2015 to 2024. It includes regulations such as reducing eco-friendly cars and carbon emissions and strengthening management around life. In the future, managing air pollution is expected to reduce children’s health effects caused by exposure to air pollution.”

--------------------------------------------------------------------------------------------

Major Comment 5

“The manuscript will need a review from a native English speaker or someone with excellent command of English.”

Response

Thank you for your comment. We agree with your suggestion. We revised the contents according to the all reviewer’s comments and entrusted the review to a person with excellent English skills.

--------------------------------------------------------------------------------------------

Reviewer 2 Report

The work proposes a time series study to evaluate the relationship between exposure to PM2.5 and acute lower respiratory infection (ALRI) hospitalizations in children (0-5 years) living in 7 metropolitan cities in Korea, considering the moving average exposure of 7 days to PM2.5 after adjusting for apparent temperature, day of the week, and hourly trends.

However, the quality of the presentation and the scientific soundness must be improved. There is a lack of information in several sections of the manuscript that must be detailed. Regarding the novelty, it remains ambiguous, since there are many recent works that use similar methods and obtain similar results. In addition, the manuscript has several typographical errors and English style problems. Comments by section are listed below.

Abstract

The abstract does not report the relevance of the work in a concrete way. An abstract should be brief and specific about the problem addressed, the methods used and the main results and conclusions of the work. Therefore, it is necessary to rewrite the abstract according to the previous recommendation.

1 Introduction

The novelty and the contribution are not clear, the proposed study must be justified according to the inconveniences or areas of opportunity of similar and recent works.

2 Methods

The methods appear adequate, although given that the time period and other variables can be arbitrary, information on the different variables used is lacking.

3 Results

The data does not seem quite right, because they refer to tables and figures that are not in the manuscript. So it is confusing or uncomfortable not being able to see that data.
The results are clear, although some interpretation of the cold season may be missing.

5 Conclusions

The conclusion is a bit short, the authors should briefly detail the highlights of the study. In addition, future work or ways to improve said study need to be addressed.

Author Response

Major Comment 1

“Abstract

The abstract does not report the relevance of the work in a concrete way. An abstract should be brief and specific about the problem addressed, the methods used and the main results and conclusions of the work. Therefore, it is necessary to rewrite the abstract according to the previous recommendation.”

Response

Thank you for your comment. We agree with your suggestion. As you have pointed out, we have concisely revised the Abstract.

After revision (In the Abstract)

“Although several studies have evaluated the association between fine particulate matter (PM2.5) and acute lower respiratory infection (ALRI) in children, their results were inconsistent. Therefore, we aimed to evaluate the association between short-term exposure to PM2.5 and ALRI hospitalizations in children (0-5 years) living in 7 metropolitan cities of Korea. The ALRI hospitalization data of children living in 7 metropolitan cities of Korea from 2008 to 2016 was acquired from a customized database constructed based on National Health Insurance data. The time-series data in a generalized additive model were used to evaluate the relationship between ALRI hospitalization and 7-day moving average PM2.5 exposure after adjusting for apparent temperature, day of the week, and time trends. We performed a meta-analysis using a two-stage design method. The estimates for each city were pooled to generate an average estimate of the associations. The average PM2.5 concentration in 7 metropolitan cities was 29.0 μg/m3 and a total of 713,588 ALRI hospitalizations were observed during the 9-year study period. A strong linear association was observed between PM2.5 and ALRI hospitalization. A 10 μg/m3 increase in the 7-day moving average of PM2.5 was associated with a 1.20% (95% CI: 0.71, 1.71) increase in ALRI hospitalization. While we found similar estimates in a stratified analysis by sex, we observed stronger estimates of the association in the warm season (1.71%, 95% CI: 0.94, 2.48) compared to the cold season (0.31%, 95% CI: −0.51, 1.13). In the two-pollutant models, the PM2.5 effect adjusted by SO2 was attenuated more than in the single pollutant model. Our results suggest a positive association between PM2.5 exposure and ALRI hospitalizations in Korean children, particularly in the warm season. The children need to refrain from going out on days when PM2.5 is high.”

--------------------------------------------------------------------------------------------

Major Comment 2

“1 Introduction

The novelty and the contribution are not clear, the proposed study must be justified according to the inconveniences or areas of opportunity of similar and recent works.”

Response (In the Introduction)

Thank you for your comment. We agree with your suggestion. We justified the need for our research in the Introduction section.

--------------------------------------------------------------------------------------------

Major Comment 3

“2 Methods

The methods appear adequate, although given that the time period and other variables can be arbitrary, information on the different variables used is lacking.”

Response 

Thank you for your comment. As you point out, the time period and other confounding may be arbitrary. We additionally adjusted weather variables such as wind and air pressure, but the results did not change.

--------------------------------------------------------------------------------------------

Major Comment 4

“3 Results

The data does not seem quite right, because they refer to tables and figures that are not in the manuscript. So it is confusing or uncomfortable not being able to see that data. The results are clear, although some interpretation of the cold season may be missing.”

Response 

Thank you for your comment. We are sorry to have caused confusion by supplemental tables and figures in appendix file. We came up with the results from various points of view, and we decided on the main table and the figures. The Content not included in the manuscript file is included in the supplementary files. However, we have transferred some content to the manuscript according to your opinion.

Before revision

  • The ALRI hospitalization was determined by the International Classification of Disease, 10th revision diagnostic codes: either primary or secondary diagnostic codes of J20-J22 (Table S1).
  • Supplementary figure 3 and 4 were moved to manuscript file.

After revision  (In the Supplementary files)

  • The ALRI hospitalization was determined by the International Classification of Disease, 10th revision diagnostic codes: either primary or secondary diagnostic codes of J20-J22 (J20: Acute bronchitis, J21: Acute bronchiolitis, J22: Unspecified acute lower respiratory infection).

The figure 4 and figure 5 have been added to the manuscript file.

--------------------------------------------------------------------------------------------

Major Comment 5

“5 Conclusions

The conclusion is a bit short, the authors should briefly detail the highlights of the study. In addition, future work or ways to improve said study need to be addressed.”

Response

Thank you for your comment. We agree with your suggestion. In the Conclusion section, we added recommendations on air pollution prevention and comments on future work.

After revision (In the Conclusion)

“Our results have revealed a link between PM2.5 and ALRI hospitalization among Korean children. The increase in PM2.5 is related to the increased ALRI hospitalization for Korean children. In particular, exposure to PM2.5 in warm seasons was highly related to ALRI. Children need to refrain from outdoor activity when PM2.5 is high. The results of our research can be used for children’s health care and health policies for children’s respiratory disease. Future research needs to investigate the link between air pollution, temperature and children's respiratory health.

Since PM2.5 is mainly caused by traffic and industrial emissions, government control is necessary. The South Korean government announced its 2nd Seoul Metropolitan Air Quality Control Master Plan from 2015 to 2024. It includes regulations such as reducing eco-friendly cars and carbon emissions and strengthening management around life. In the future, managing air pollution is expected to reduce children’s health effects caused by exposure to air pollution.”

-------------------------------------------------------------------------------------------

Reviewer 3 Report

The paper analyses the relationship between short term (7 day) exposure to PM2.5 and acute lower respiratory infection (ALRI) in children and found, hat there is a positive correlation between PM2.5 and ALRI, the realtion is larger in summer.

A weaker point is the choice of the data for the PM2.5 concentration used in the study. The results of an atmospheric model (CMAQ) were taken. Even the best models are not very good when estimating PM2.5 and PM10 concentrations. Often, modelled data is ‘adjusted’ with measured data, but whether this has been done within this study is not mentioned.

Furthermore, urban concentrations of PM10 and PM2.5 are larger than rural concentrations, and also the variation of the concentration is larger in cities. Values are calculated for 9km*9km grids, but is this sufficient to get urban averaged concentrations? This point should be adressed

With regard to daily variations of concentrations: the daily emissions are highly temperature dependent – has this been taken into account in the emission preprocessor of the model or have the modellers just used seasonal split factors for the temporal resolution. ?

I assume that you used modelled and not measured data, as at least for the earlier years no PM2.5 measurements have been available? However, parts of the diseases analysed (e.g. bronchitis) might be correlated with PM10. So it might have been interesting to correlate measured urban background PM10 with ALRI. Why has this not been done?

That effects of PM2.5 exposure are higher with higher temperature, is also observed – however not for ALRI in children, but other diseases from PM2.5 exposure by

Lepeule, J., Litonjua, A.A., et al., 2018. Lung function association with outdoor temperature and relative humidity and its interaction with air pollution in the elderly, Environmental Research, 165, 110-117. https://doi.org/10.1016/j.envres.2018.03.039.

or  Li, J., et al., Modification of the effects of air pollutants on mortality by temperature: A systematic review and metaanalysis, Sci Total Environ (2016), http://dx.doi.org/10.1016/j.scitotenv.2016.10.070

These findings could be included in the discussion.

In summary, it is valuable to have a further study confirming a relation between PM2.5 and acute lower respiratory infection (ALRI) in children. Limitations have been addressed, however I propose to also include and discuss the issues mentioned above.

corrections:

line 63: research showed no relationship

line 88: uses ... follows

table 1, line 176: Daejeon warm season: 16,566

page 7 line 12: All models used...; same for Table 3.

page numbered as 3 (though it is not page 3), line 10-12: ‘Another nationwide study in Korea in children aged 0-4 reported a null association between PM2.5 and ALRI hospitalization [21], but a significant association in the cold season.’? I do not understand this sentence.

Author Response

Major Comment 1

“The paper analyses the relationship between short term (7 day) exposure to PM2.5 and acute lower respiratory infection (ALRI) in children and found, hat there is a positive correlation between PM2.5 and ALRI, the realtion is larger in summer.

A weaker point is the choice of the data for the PM2.5 concentration used in the study. The results of an atmospheric model (CMAQ) were taken. Even the best models are not very good when estimating PM2.5 and PM10 concentrations. Often, modelled data is ‘adjusted’ with measured data, but whether this has been done within this study is not mentioned. Furthermore, urban concentrations of PM10 and PM2.5 are larger than rural concentrations, and also the variation of the concentration is larger in cities. Values are calculated for 9km*9km grids, but is this sufficient to get urban averaged concentrations? This point should be addressed. With regard to daily variations of concentrations: the daily emissions are highly temperature dependent – has this been taken into account in the emission preprocessor of the model or have the modellers just used seasonal split factors for the temporal resolution. ?”

Response

Thank you for your comment. We agree with your concern about our exposure assessment. As you are concerned, we are careful in using PM2.5 modeling data.

Therefore, we were consulted by exposure experts among co-authors (Prof. Soontae Kim is an atmosphere expert who majored in environmental engineering). Through consultation with the co-authors, we have reduced the concerns about exposure assessment in the following ways.

After revision (In the Supplementary files)

“Daily PM2.5 concentration was estimated with meteorology inputs simulated from a Weather Research and Forecasting (WRF, version 3.3.1) model and anthropogenic emissions processed through the Sparse Matrix Operator Kernel Emission (SMOKE, version 3.1) [9-11]. For the early stages of weather conditions, National Center for Environmental Prediction/Final Analysis (NCEP-FNL), a reanalysis data from the U.S. Oceanic and Atmospheric Administration (NOAA), was used. The Model of Emissions of Gases and Aerosols from Nature (MEGAN, version 2.04) was used to estimate emissions of biogenic volatile organic compounds. To simulate the secondary aerosol formation and growth, and gas-phase chemical reactions, the 5th generation CMAQ Aerosol Module (AERO5) and the statewide air pollution research center version 99 (SAPRC99) were used for the aerosol and chemical mechanisms, respectively [9-11]. For air quality simulation, two model domains were used for further analysis, with horizontal grid resolutions of 27 km (covering Northeast Asia) to include regional transports of PM2.5 and its precursors, and 9 km (South Korea) to prepare the nationwide PM2.5 at a finer grid resolution. Having the air quality simulation platform ready, we first simulated the hourly concentration of PM2.5 at each grid and resampled the values to the Earth level based on the GIS shape-file of Korea [12]. Our exposure data are data of spatial interpolation results based on observational data. The observational data are monitored data provided by the Ministry of Environment. We considered factors that affect atmospheric concentrations such as emissions, weather condition, and topography for space interpolation. But, there are some missing points in the spatial and temporal aspects of the observational data. In particular, South Korea has many coastal and islands because it is a peninsula surrounded by the sea. In coastal or islands, there is no data to interpolate. Instead, we used the simulation results. We’ve interpolated the space by the hour. Domestic emission data used Clean Air Policy Support System (CAPSS) 2010 and overseas emission data used MICS-Asia 2010 (Model Inter-Comparison Study for Asia). CAPSS is a list of annual national emissions provided by the National Institute of Environmental Research. In conclusion, we used the exposure data fused with the WRF and CMAQ simulation based on the observational data.”

-------------------------------------------------------------------------------------------

Major Comment 2

“I assume that you used modelled and not measured data, as at least for the earlier years no PM2.5 measurements have been available? However, parts of the diseases analysed (e.g. bronchitis) might be correlated with PM10. So it might have been interesting to correlate measured urban background PM10 with ALRI. Why has this not been done?”

Response

Thank you for your comment. We agree with your suggestion. We analyzed the association between ALRI hospitalization with PM10 data measured by monitoring. Like PM2.5, PM10 shows a similar association to ALRI hospitalization, but the effect size is slightly smaller. However, we tried to focus on the relationship between PM2.5 and ALRI hospitalization in this study. The results of the analysis of the association between PM10 and ALRI hospitalization have been added to the below table.

PM2.5

PM10

Cities

Percentage change (95% CI)

Percentage change (95% CI)

Seoul

1.60 (1.00, 2.20)

1.28 (0.91, 1.65)

Busan

0.81 (0.17, 1.45)

0.68 (0.32, 1.03)

Daegu

1.64 (0.70, 2.58)

1.14 (0.60, 1.68)

Incheon

0.57 (-0.45, 1.61)

0.90 (0.30, 1.50)

Gwangju

0.28 (-0.51, 1.08)

0.08 (-0.37, 0.53)

Daejeon

2.09 (0.83, 3.37)

1.38 (0.49, 2.28)

Ulsan

1.92 (0.71, 3.15)

0.93 (0.23, 1.64)

Random effect

1.20 (0.71, 1.71)

0.87 (0.55, 1.19)

--------------------------------------------------------------------------------------

Major Comment 3

“That effects of PM2.5 exposure are higher with higher temperature, is also observed – however not for ALRI in children, but other diseases from PM2.5 exposure by

Lepeule, J., Litonjua, A.A., et al., 2018. Lung function association with outdoor temperature and relative humidity and its interaction with air pollution in the elderly, Environmental Research, 165, 110-117. https://doi.org/10.1016/j.envres.2018.03.039.

or  Li, J., et al., Modification of the effects of air pollutants on mortality by temperature: A systematic review and metaanalysis, Sci Total Environ(2016), http://dx.doi.org/10.1016/j.scitotenv.2016.10.070

These findings could be included in the discussion. In summary, it is valuable to have a further study confirming a relation between PM2.5 and acute lower respiratory infection (ALRI) in children. Limitations have been addressed, however I propose to also include and discuss the issues mentioned above.”

Response

Thank you for your comment. We agree with your suggestion. We reviewed your suggested paper and added it to the Discussion section. Also, we feel the need for further study. The future research will look at the association between air pollution and temperature.

Before revision

None

After revision (In the Discussion section)

" Also, exposure to high temperatures and fine dust can affect the lung function of the elderly as well as in ALRI on children [13, 14]. The interaction of high temperatures and PM2.5 may be the cause, and these effects need to be addressed in future studies.”

------------------------------------------------------------------------------------------

Minor Comment 1 (corrections)

“line 63: research showed no relationship”

Response

Thank you for your comment. We agree with your suggestion. We revised the previous sentences as follows.

Before revision

“However, a recent Korean research a presented a no relationship between PM2.5 and ALRI when the authors investigated nationwide childhood respiratory hospitalization in 2013-2015”

After revision

“However, a recent Korean research showed no relationship between PM2.5 and nationwide childhood respiratory hospitalization in 2013-2015”

-------------------------------------------------------------------------------------

Minor Comment 2 (corrections)

line 88: uses ... follows

Response

Thank you for your comment. We agree with your suggestion. We revised the previous sentences as follows.

Before revision

“This study use the unidentifiable data prepared by the NHIS and follow the guidelines for the protection of personal information of the NHIS.”

After revision (In the Introduction section)

“This study uses the unidentifiable data prepared by the NHIS and follows the guidelines for the protection of personal information provided by the NHIS.”

----------------------------------------------------------------------------------------

Minor Comment 3 (corrections)

table 1, line 176: Daejeon warm season: 16,566

Response

Thank you for your comment. We revised the contents of the table you pointed out.

Before revision

“165,66”

After revision (In the table 1)

“16,566”

------------------------------------------------------------------------------------------

Minor Comment 4 (corrections)

page 7 line 12: All models used...; same for Table 3.

Response

Thank you for your comment. We revised the sentences as follows.

Before revision

“All models were used the 7-day moving average exposure to PM2.5 μg/m3.”

After revision (In the Results section)

“All models used the 7-day moving average exposure to PM2.5 μg/m3.”

------------------------------------------------------------------------------------------

Minor Comment 5 (corrections)

page numbered as 3 (though it is not page 3), line 10-12: ‘Another nationwide study in Korea in children aged 0-4 reported a null association between PM2.5 and ALRI hospitalization [21], but a significant association in the cold season.’? I do not understand this sentence.

Response

Thank you for your comment. We have revised the sentences to avoid confusion.

Before revision

“Another nationwide study in Korea in children aged 0-4 reported a null association between PM2.5 and ALRI hospitalization [21], but a significant association in the cold season.”

After revision (In the Discussion section)

“Another nationwide study in Korea in children aged 0-4 reported a stronger association between PM2.5 and ALRI hospitalization in the cold season [21]”